# DagSim: Combining DAG-based model structure with unconstrained data types and relations for flexible, transparent, and modularized data simulation

Ghadi S. Al Hajj[1]*, Johan Pensar[2], Geir K. Sandve[1]

1 Department of Informatics, University of Oslo, Oslo, Norway, 2 Department of Mathematics, University of Oslo, Oslo, Norway

* ghadia@uio.no, ghadi.alhajj8@gmail.com

**Data Availability Statement:** The data used in this paper are generated using computer simulations. The reader can simulate similar data using the described tool, which is available as a python

## Abstract

Data simulation is fundamental for machine learning and causal inference, as it allows exploration of scenarios and assessment of methods in settings with full control of ground truth. Directed acyclic graphs (DAGs) are well established for encoding the dependence structure over a collection of variables in both inference and simulation settings. However, while modern machine learning is applied to data of an increasingly complex nature, DAG-based simulation frameworks are still confined to settings with relatively simple variable types and functional forms. We here present DagSim, a Python-based framework for DAG-based data simulation without any constraints on variable types or functional relations. A succinct YAML format for defining the simulation model structure promotes transparency, while separate user-provided functions for generating each variable based on its parents ensure simulation code modularization. We illustrate the capabilities of DagSim through use cases where metadata variables control shapes in an image and patterns in bio-sequences. DagSim is available as a Python package at PyPI. Source code and documentation are available at: https://github.com/uio-bmi/dagsim

## Introduction

Data simulation is fundamental for machine learning (ML) and causal inference (CI), as it allows ML/CI methods to be evaluated in a controlled setting using a ground truth model [1–3]. For the purpose of designing flexible, controllable, and transparent simulator models, the class of directed acyclic graphs (DAGs) provides a highly useful framework. The DAG is used to encode the structure of a model involving multiple variables in a form that is both compact and intuitive to a human user. In addition, the resulting DAG-based model is modular and allows for building complex simulators from simpler local components or modules. In a purely probabilistic model, known as a Bayesian Network (BN) [4], the DAG is used to specify the dependence structure over the considered variables. In a causal model, known as a structural causal model (SCM) [5], the DAG is used to specify the causal structure of the underlying

package (https://pypi.org/project/dagsim/). The code (available as python scripts and as jupyter notebooks) for simulating the data used in the manuscript usecases can be found on GitHub (https://github.com/uio-bmi/dagsim/tree/main/manuscript_usecases). Also, the user can run these simulations online by clicking on the "launch binder" badge provided on that page, using the free Binder service.

**Funding:** The author(s) received no specific funding for this work.

**Competing interests:** The authors have declared that no competing interests exist.

data-generating process. In either case, a simulation model is defined by specifying the functional relations between each node and its parents in the assumed graph. In a BN, these relations are typically defined as probability distributions, while an SCM typically models relations as deterministic, where the value of a node is computed based on the value of its parents and an additional exogenous random variable (often referred to as a noise variable). In terms of simulation, there is no practical distinction between the purely probabilistic (BN) and causal perspective (SCM)–in either case, data is generated through direct forward sampling following a node ordering that is consistent with the given DAG (known as a topological ordering). However, the fundamental difference is that an SCM is equipped with some additional causal capabilities that go beyond those of a BN. For example, scenarios involving interventions and counterfactuals can be simulated by making simple local modifications to the original model ahead of a standard simulation.

While there is in principle no limitation on the types of variables or functional forms in the BN and SCM frameworks, the main emphasis has historically been on relatively small DAGs with variables of basic data types (typically ordinal/categorical scalar values) [6–12]. A visual notation known as plate notation is well-established for denoting vector-valued variables in BNs, but representing a k-dimensional tensor requires a fixed k-level nesting of plates, and there is no well-established notation for representing sets or sequences. This stands in stark contrast to the recent neural network (NN)-driven machine learning revolution, where the main aspect has been the ability to learn useful representations from data of large dimensionality and complex structure [13–15]. The canonical example of this is the learning of complex signals from large, two-dimensional structures of pixel values in image analysis, as well as from sequences of words in natural language processing.

The emphasis on simple types of variables and functional relations in the graphical model field is also apparent from the programming libraries available for structure learning, parameter inference and simulation from graphical models. For example, the seminal bnlearn R package [9] can both infer parameters and simulate data from a model but is restricted to numerical variables, whether discrete or continuous, and restricted to full conditional probability tables and linear regression models as functional forms. DAG-based simulation is also supported in a variety of other packages, either as the main purpose or as a side purpose (their properties in terms of simulation are summarised in Table 1). Many of the packages are explicitly restricted to linear relations, as in the structural equation models (SEM) framework. All the mentioned packages share with bnlearn the restriction to numerical variables and particular functional forms.

We here argue for the usefulness of combining the ideas of carefully designed models of variable relations from the graphical modelling field with the complex data types that are characteristic of the current wave of NN-driven deep learning. We present a Python library DagSim that streamlines the specification of simulation scenarios based on graphical models where variables and functional relations can be of any form. The fundamental idea of the framework is simple yet powerful: allowing the user to define a DAG-based simulation by connecting nodes to their parents through standard Python functions with unrestricted parameter and return values. DagSim provides a range of functionality for increasing the convenience and transparency of such a simulation setup—offering a choice between an external (YAML-based) or internal (Python-based) succinct and transparent domain-specific language (DSL) to specify simulations involving plates, mixture distributions and various missing data schemes. It also includes functionalities of specific use for the simulation of causal scenarios, including native support for simulating sample selection bias.

**Table 1. An overview of all established frameworks that to the authors' knowledge offer DAG-based simulation functionalities, describing for each package the main purpose, the type of data it simulates, the functional forms used, and the additional simulation utilities provided.** The bnlearn package [9] can both infer parameters and simulate data from a model, with numerical variables and functional forms restricted to full conditional probability tables and linear regression models. The pgmpy package [16] is similar to bnlearn in terms of its purpose and simulation functionalities. The package simCausal [11] is more aimed toward causal inference problems and thus focuses on simulating longitudinal data based on SEMs. The main goal of the simMixedDAG package [12] is to simulate "real life" datasets based on a learned generalised additive model or user-defined parametric linear models. The package MXM [7] simulates data from multivariate Gaussian distributions based on a user-defined or randomly generated adjacency matrix, while abn [6] simulates data from Poisson, multinomial, and Gaussian distributions based on a user-defined adjacency matrix. The packages dagitty [10], dagR [17], and lavaan [8] provide similar functionalities for simulating data based on SEMs.

| Framework | Main Purpose | Data type | Functional form | Distinctive features | Reference |
|---|---|---|---|---|---|
| DagSim | Data simulation | Any data type (passed directly between nodes) | Any form (custom function) | Plates, selection bias, missing values, stratification | This paper |
| bnlearn | Structure, and parameter learning | Discrete and continuous | Categorical distribution and linear Gaussian model | Includes several off-the-shelf models | [9] |
| pgmpy | Model learning, and approximate, exact, and causal inference | Discrete and continuous | Categorical distribution and linear Gaussian model | Includes several off-the-shelf models | [16] |
| simCausal | Simulation of SEM-based complex longitudinal data structures | Discrete and continuous | Linear model | Counterfactual data, interventions, time-varying nodes | [11] |
| simMixedDAG | Simulation of data from parametric and non-parametric DAG models | Discrete and continuous | Generalized additive model | Learns a non-parametric model from data | [12] |
| MXM | Feature selection | Discrete and continuous | Linear Gaussian model | Simulates a DAG with arbitrary arc density | [7] |
| abn | Modelling data with additive Bayesian networks | Discrete and continuous | Generalized linear model | Simulates a DAG with arbitrary arc density | [6] |
| dagitty | Graphical analysis of Structural Causal Models | Binary and continuous | Linear Gaussian and logistic models | Characterisation, restructuring and random generation of DAGs | [10] |
| dagR | Construct and evaluate DAGs, and simulate data | Binary and continuous | Linear Gaussian and logistic models | Includes several off-the-shelf models | [17] |
| lavaan | Latent Variable Analysis | Continuous | Linear model | Fits a latent variable model to data | [8] |

## Implementation

To specify a DagSim simulation model, a user simply defines a set of nodes (variables) along with possible input nodes (parents), which together make up the backbone of the model in the form of a directed graph. The user then assigns a general Python function for simulating the value of each node given the values of parent nodes, if any. When the model has been fully specified, the package checks that the provided directed graph is acyclic, thus ensuring that the values of any input node can be sampled prior to all downstream nodes. Following a topological ordering of the nodes, DagSim then uses standard forward sampling to simulate the values of each node by calling its assigned function and providing the values of its parents (if any) as arguments. Importantly, parent values are directly passed on as native Python values, ensuring that the framework supports general data types and any functional forms. The simulated data is saved to a CSV file.

In addition to more standard simulation scenarios, DagSim provides additional types of nodes that facilitate the simulation of different types of real-world scenarios. The Selection node, for example, allows the user to simulate selection bias [5] through a function that governs the sample selection scheme, where the arguments for that function are similar to those of a standard node. The Missing node, on the other hand, provides a convenient way to simulate missing entries in the data by specifying the variable that will consequently have missing values and another standard node that specifies which entries should be removed. Finally, the Stratify node offers a way to automatically stratify the resulting data into separate files through a single function that defines the stratum of each sample. Additionally, DagSim supports transparent a specification of simulations based on a succinct YAML format.

## Step-by-step example

Suppose you would want to simulate sequences of coin tosses, each represented as a 10–20 long text of H (head) and T (tail), according to a sample-specific probability of getting heads, sampled itself from a uniform distribution. Fig 1 shows the overall workflow one would follow:

- First, define one node for the probability of getting Heads, one node for the number of coin tosses per sample, and one node for the sampled sequence itself (which has incoming edges from the two other nodes), using e.g. YAML to specify the graph

- Second, define the simulation instructions in the YAML file

- Third, define the custom function for simulating a sequence of tosses, with the other two nodes utilizing already existing functions, e.g. using numpy.

- Finally, simply run the defined simulation, e.g. from the command line.

The code and files corresponding to this example can be found in the supplementary material.

## Use

The driving motivation for DagSim is the ability to combine basic (scalar, ordinal/categorical) variables with complex or high-dimensional data types in a transparent structure as defined by a DAG, without any restrictions on the complexity of the functional relations between nodes. We illustrate these capabilities through two stylized simulation examples, where basic meta-data variables are controlling 1) shapes in an image (two-dimensional numeric tensor), and 2) bio-sequence patterns in an adaptive immune receptor repertoire (set of sequences). Our main

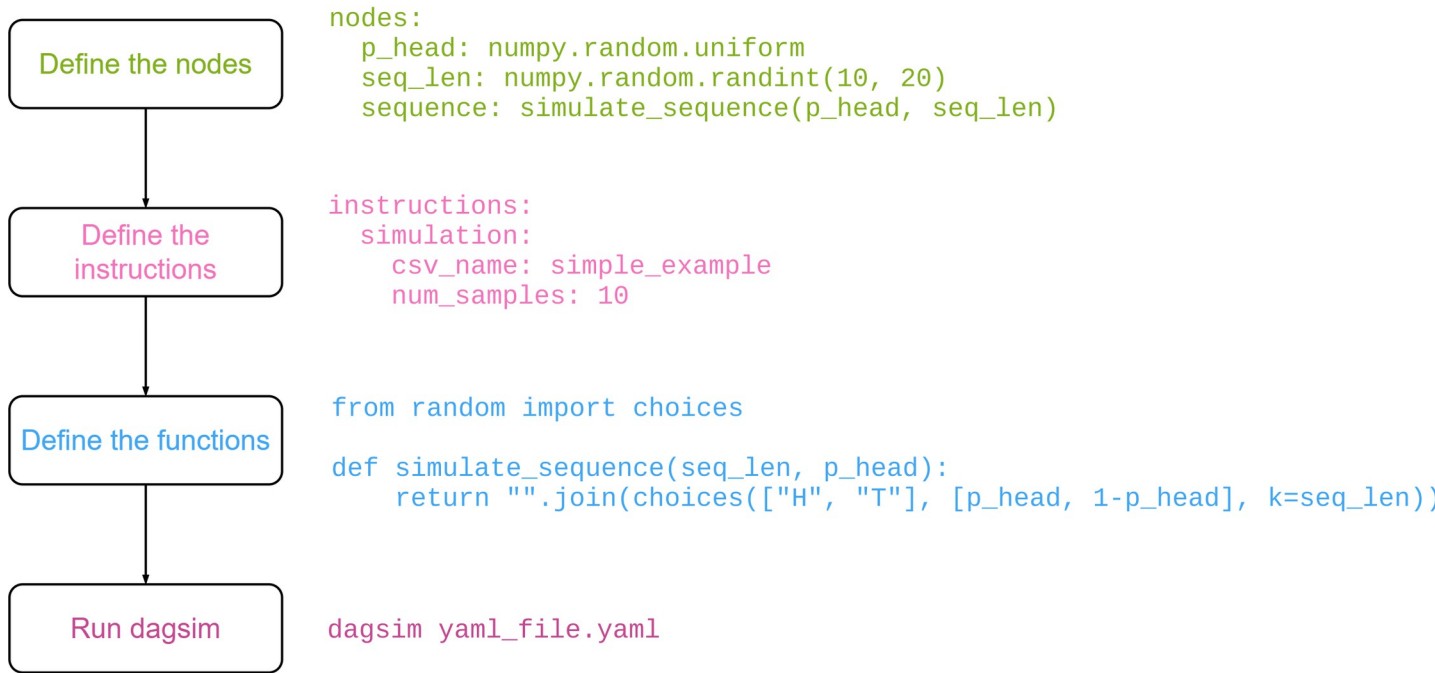

**Fig 1. A typical workflow of simulating data using DagSim.**

emphasis is on the ease of defining simulations and the transparency of the resulting simulation models. Detailed versions of these examples can be found in S1 and S2 Figs, respectively.

A first use case is based on a study by Sani et. al. [18] on causal learning as a tool for explaining the behaviour of black-box prediction algorithms. In order to illustrate their approach, they simulated simple images with specific shapes overlaid on a black background, where an additional set of scalar variables controlled the probability of each of the different shapes being introduced to the image. We show how such a simulation is easily reproduced using DagSim, based on a succinct, transparent, and well-modularized model specification (Fig 2A and 2B). The simulation of each node given its parents is defined by a set of Python functions provided

```
1  graph:
2    nodes:
3      U1: uniform(0,1)
4      U2: uniform(0,1)
5      H: binomial(1, U1)
6      C: binomial(1, U2)
7      V: complement_binomial(U1)
8      R: sigmoid_binomial(C, H, "H")
9      Y: sigmoid_binomial(C, V, "V")
10     Image: drawImage(H, V, R, C)
11   python_file: ImagesExample.py
12
13 instructions:
14   simulation:
15     csv_name: Images_metadata
16     num_samples: 50
```

(a)

(b)

```
1  graph:
2    nodes:
3      Disease: binomial(1, 0.5)
4      Age: randint(10,80)
5      Protocol: assign_protocol(Disease)
6      AIRR:
7        function: create_airr(Disease, Age, Protocol)
8        observed: False
9      kmerVec: encode_kmers(AIRR)
10   python_file: BioseqExample.py
11 instructions:
12   simulation:
13     num_samples: 50
14     csv_name: "BioseqExample_yaml"
```

(c)

(d)

**Fig 2.** (a-b) The YAML specification and corresponding DAG for the image simulation use case, (c-d) The YAML specification and corresponding DAG for the biosequence simulation use case.

in the supplementary material, where the main function is the one that generates an image conditioned on the scalar metadata values. If this use case was to be performed by any of the existing DAG-based simulation frameworks, then the scalar values would have to be simulated separately using an appropriate DAG. Following that, a separate function that takes as input the variables V, C, R, and H and iterates on all the samples could be used to create the desired image. This would detach the image construction process from the rest of the simulation making the code unnecessarily complicated and less transparent.

A second use case exemplifies simulation in settings of high-dimensional biomarkers and low-dimensional patient characteristics. The considered biomarker is based on sequence patterns in a gene known as the immune receptor, which is reflecting what individual adaptive immune cells are recognizing and reacting to. The set of DNA sequences for this gene across all adaptive immune cells in the body is collectively known as the adaptive immune receptor repertoire (AIRR). Any disease state with immune system involvement, including infectious disease, auto-immunity and cancer, introduces sequence patterns in the AIRR. Additionally, it has been proposed that immune repertoires become less diverse with age [19] and that experimental protocols introduce particular sequence biases in observed AIRR datasets [20, 21]. Simulation of such biomarker signals allows benchmarking the ability of the current methodology to infer disease state from AIRR patterns [22, 23], as well as to assess the robustness of the learning process to variability in patient characteristics and experimental biases [24]. The model specification and resulting DAG are shown in Fig 2C and 2D. The Python functions are provided in the supplementary material, where the main function simulates the AIRR for each patient conditioned on disease state, age, and the experimental protocol used. If this use case was to be performed by any of the existing DAG-based simulation frameworks, one would have to use a numeric representation of the sequences, with for example the use of ad hoc end-of-sequence numeric codes to emulate a set of variable length sequences. As the frameworks also do not support the specification of custom functions, one would need to supplement a DAG-based simulation of baseline sequences (in numeric representation) with post-hoc functions for implanting desired signals.

## Conclusion

We have here argued that DAG-based simulation should transcend the traditional settings of only numeric-valued variables, allowing the convenience and transparency of graphical models to see use also in settings with more complex data types. Specifically, complex data types would bring these simulations closer to applications typically considered by modern machine learning (often deep learning) models. Hence, we consider the integration of complex data types and graphical modelling for simulation purposes as highly timely, given both the increasing inclusion of complex data types in modelling scenarios and the increasing interest in causal concepts in the ML field. In terms of the latter, there has been recent research into how underlying causal mechanisms affect ML strategies [25], research into how the underlying causal structure determines whether data from different sources can be successfully fused for learning ML models [26], research into how overlaid signals arising from distinct mechanism can be disentangled [27, 28], calls for extending modern ML methods to directly predict effects of interventions [29], calls for incorporating non-linear machine learning methods for causal inference in epidemiology [30], and an increasing interest in how causal mechanisms affect the stability (generalizability) of ML models when applied to new settings [24, 31]. The combination of a DAG-based model backbone with flexible data types and functional relations provides for transparent and modularized simulation models in these emerging settings, where low-dimensional variables are connected to complex patterns in high-dimensional variables.

Through a succinct YAML format for defining the model backbone and the use of individual, native Python functions for defining the functional relations to each node, DagSim provides a straightforward, practical implementation to support such an approach. The framework also natively supports specific functionalities that are useful when simulating data, e.g. for emulating selection bias and missing data, and could in the future be further extended to natively support features like Dynamic Bayesian Network-based simulation of time series, as well as nested and intersecting plates structures for complex modelling scenarios [4]. More important than individual features is the overall ability to exploit DAG structures to improve transparency and code modularization. The examples of simulating shapes in images and patterns in biosequences are but two exemplifications of DagSim's advantages. While existing frameworks also allow to define transparent simulations in settings with standard functional relations between numeric scalars or vectors, only DagSim allows transparency and code modularization in a broad range of settings with complex data types and functional relations.

## Supporting information

**S1 Fig. DAG for use case I.**
(TIF)

**S2 Fig. DAG for use case II.**
(TIF)

## Acknowledgments

We wish to thank Victor Greiff and Anne H Schistad Solberg for their input on the manuscript text, and we wish to thank Milena Pavlović and Knut Rand for feedback after trying out the software.

## Author Contributions

**Conceptualization:** Ghadi S. Al Hajj, Johan Pensar, Geir K. Sandve.

**Software:** Ghadi S. Al Hajj.

**Supervision:** Johan Pensar, Geir K. Sandve.

**Writing – original draft:** Ghadi S. Al Hajj, Geir K. Sandve.

**Writing – review & editing:** Ghadi S. Al Hajj, Johan Pensar, Geir K. Sandve.

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
