## [Decision Letter · Decision Letter 0]

26 Jan 2023

PONE-D-22-32989DagSim: Combining DAG-based model structure with unconstrained data types and relations for flexible, transparent, and modularized data simulationPLOS ONE

Dear Dr. Al Hajj,

Thank you for submitting your manuscript to PLOS ONE. After careful consideration, we feel that it has merit but does not fully meet PLOS ONE’s publication criteria as it currently stands. Therefore, we invite you to submit a revised version of the manuscript that addresses the points raised during the review process.

We look forward to receiving your revised manuscript.

Kind regards,

Emmanuel S Abador

Academic Editor

PLOS ONE

Journal Requirements:

2. ‘Please include your tables as part of your main manuscript and remove the individual files. Please note that supplementary tables (should remain/ be uploaded) as separate "supporting information" files.’

Additional Editor Comments:

(1) There should be description section that presents step-by-step development and also the use of a flow chart is encouraged;

(2) Supplementary Material 2 is not available

Reviewers' comments:

Reviewer's Responses to Questions

**Comments to the Author**

1. Is the manuscript technically sound, and do the data support the conclusions?

Reviewer #1: Yes

Reviewer #2: Yes

2. Has the statistical analysis been performed appropriately and rigorously? 

Reviewer #1: N/A

Reviewer #2: N/A

3. Have the authors made all data underlying the findings in their manuscript fully available?

Reviewer #1: Yes

Reviewer #2: Yes

4. Is the manuscript presented in an intelligible fashion and written in standard English?

Reviewer #1: Yes

Reviewer #2: Yes

5. Review Comments to the Author

Reviewer #1: In the manuscript “DagSim: Combining DAG-based model structure with unconstrained data types and relations for flexible, transparent, and modularized data simulation”, Ghadi et al. present a python based framework for DAG-based data simulations: DagSim which integrates complex data types and graphical modelling (that is, neither restricted by variable types nor functional forms). More so, the simulation model is defined in the YAML format promoting transparency. The authors demonstrated the framework on two use cases which are available on Github (examples, where basic metadata variables are controlling the shapes in an image and patterns in bio-sequences).

Major comments:

Table 1: An overview of the current frameworks that offer simulation functionalities. The authors provided detailed information showing how these tools/frameworks differ.

1. However, they are not clear on the selection criteria they used for the presented current frameworks (are these all the existing tools/packages?)

2. When simulating simple DAGs (discrete and continuous data), how does dagSim compare to the other simulation frameworks (for example, simMixedDAG and dagR)?

Minor comment:

Table 1: An overview of the current frameworks that offer simulation functionalities.

• dagR does not have any simulation utilities, however, Duan et al. 2021 (Reflection on modern methods: understanding bias and data analytical strategies through DAG-based data simulations, https://academic.oup.com/ije/article/50/6/2091/6272915) demonstrated that dagR can be useful in addressing selection and information bias analysis. Please, clarify if dagitty and dagR doesn’t have simulation utilities

Reviewer #2: The authors introduce a novel Direct Acyclic Graph (DAG) - based data simulation Python library called DagSim. They survey existing data simulation tools, discussing them and highlighting a common limitation i.e. that their inputs are limited to numeric variables and specified functional forms. DagSim has the unique advantage of having the capacity to use variables and functional relations of any form. Furthermore, it is modular and easily updatable. An implementation of DagSim has been made available on Github.

The survey is instructive and potentially a worthy contribution. I would suggest the following modifications:

-Table 1, as provided, is a very useful summary of existing tools, their capabilities, and limitations. I would recommend an additional column in which associated citations for the tools are provided.

-In addition to Figure 1, it would be useful to present a demonstration of an example of the limitation of the other tools’ input variable limitations, and DagSim’s advantage in processing more complex data types.

Minor edits:

-pages need to be numbered

-there is a citation in the Abstract. That needs to be removed.

-In the last section under “Use”, “Additionally, it is known that e.g. the age of a patient leaves a mark on the AIRR (19)…” needs to be corrected

6. PLOS authors have the option to publish the peer review history of their article (what does this mean?). If published, this will include your full peer review and any attached files.

Reviewer #1: No

Reviewer #2: No

---

## [Author Response · Author response to Decision Letter 0]

16 Feb 2023

We would like to thank the editor and the reviewers for their helpful comments. Please find the response in the file named "Response to Reviewers".

---

## [Decision Letter · Decision Letter 1]

30 Mar 2023

DagSim: Combining DAG-based model structure with unconstrained data types and relations for flexible, transparent, and modularized data simulation

PONE-D-22-32989R1

Dear Dr. Al Hajj,

We’re pleased to inform you that your manuscript has been judged scientifically suitable for publication and will be formally accepted for publication once it meets all outstanding technical requirements.

Kind regards,

Emmanuel S Adabor

Academic Editor

PLOS ONE

Additional Editor Comments (optional):

Reviewers' comments:

Reviewer's Responses to Questions

**Comments to the Author**

1. If the authors have adequately addressed your comments raised in a previous round of review and you feel that this manuscript is now acceptable for publication, you may indicate that here to bypass the “Comments to the Author” section, enter your conflict of interest statement in the “Confidential to Editor” section, and submit your "Accept" recommendation.

Reviewer #1: All comments have been addressed

Reviewer #2: All comments have been addressed

2. Is the manuscript technically sound, and do the data support the conclusions?

Reviewer #1: Yes

Reviewer #2: Yes

3. Has the statistical analysis been performed appropriately and rigorously? 

Reviewer #1: N/A

Reviewer #2: N/A

4. Have the authors made all data underlying the findings in their manuscript fully available?

Reviewer #1: Yes

Reviewer #2: Yes

5. Is the manuscript presented in an intelligible fashion and written in standard English?

Reviewer #1: Yes

Reviewer #2: Yes

6. Review Comments to the Author

Reviewer #1: (No Response)

Reviewer #2: (No Response)

7. PLOS authors have the option to publish the peer review history of their article (what does this mean?). If published, this will include your full peer review and any attached files.

Reviewer #1: No

Reviewer #2: **Yes: **George K. Acquaah-Mensah

---

## [Editor Report · Acceptance letter]

5 Apr 2023

PONE-D-22-32989R1 

DagSim: Combining DAG-based model structure with unconstrained data types and relations for flexible, transparent, and modularized data simulation 

Dear Dr. Al Hajj:

I'm pleased to inform you that your manuscript has been deemed suitable for publication in PLOS ONE. Congratulations! Your manuscript is now with our production department. 

Kind regards, 

on behalf of

Dr. Emmanuel S Adabor 

Academic Editor

PLOS ONE